# Design of Multi-Stage Roll Die Forming Process for Drum Clutch with Artificial Neural Network

**DOI:** 10.3390/ma14010069

**Published:** 2020-12-25

**Authors:** Jae-Hong Kim, Jae-Chang Ryu, Woo-Sik Jang, Joon-Hong Park, Young-Hoon Moon, Dae-Cheol Ko

**Affiliations:** 1Engineering Research Center of Innovative Technology on Advanced Forming, Pusan National University, 2, Busandaehak-ro 63beon-gil, Geumjeong-gu, Busan 46241, Korea; kjh86@pusan.ac.kr (J.-H.K.); yhmoon@pusan.ac.kr (Y.-H.M.); 2Department of Nanomechatronics Engineering, Pusan National University, 2, Busandaehak-ro 63beon-gil, Geumjeong-gu, Busan 46241, Korea; jcr0416@pusan.ac.kr (J.-C.R.); petitpe86@naver.com (W.-S.J.); 3Department of Mechanical Engineering, Dong-A University, 37, Nakdong-daero 550beon-gil, Saha-gu, Busan 49315, Korea; acttom@dau.ac.kr

**Keywords:** multi-stage roll die forming, drum clutch, dimensional accuracy, artificial neural network

## Abstract

The multi-stage roll die forming (RDF) process is a plastic forming process that can manufacture a transmission part with a complex shape, such as a drum clutch, by using a die set with rotational rolls. However, it is difficult to satisfy dimensional accuracy because of spring-back and unfilling. The purpose of this study is to design a multi-stage RDF process for the manufacturing of a drum clutch to improve dimensional accuracy using an artificial neural network (ANN). Finite element (FE) simulation of the multi-stage RDF process is performed to predict the dimensional accuracy according to various clearances for each stage. Moreover, the ANN is used to determine the relationship between the clearance and dimensional accuracy of the drum clutch to reduce the number of FE simulation. The results of the FE simulation and ANN are used to determine the optimal clearance for each stage of the RDF process. Finally, the drum clutch is fabricated using the determined conditions. The experimental results are in good agreement with the results of FE simulation from the aspect of outer diameter, inner diameter, thickness of outer tooth, thickness of inner tooth, and face thickness of tooth.

## 1. Introduction

Recently, power loss reduction and the application of lightweight components have been increasing fuel efficiency in the automotive industry to comply with strict environmental regulations [1,2]. Especially, the multi-stage automotive transmission system has been investigated in order to improve fuel efficiency. The multi-stage automotive transmission system includes many components, such as clutches, brakes, and gears. A drum clutch, which is the main component in automotive transmission systems, must satisfy minimum requirements concerning both strength and dimensional accuracy.

The components used in automotive transmission systems are generally manufactured by machining process or various cold forming processes, such as deep drawing, forging, groove threading, upsetting, and ironing [3,4,5,6,7]. The machining process leads to a decrease in tooth strength of components because a grain flow is cut by a shaving and hobbing process [3]. In the case of the cold forming process, the life of tools can be shortened by large forming load and productivity is low owing to complex manufacturing procedure [8,9]. Therefore, the application of effective manufacturing processes is required to prevent decrease in tooth strength and improve the productivity of the drum clutch.

The multi-stage roll die forming (RDF) has been introduced to manufacture a drum clutch, of which a tooth is formed by a successive drawing process using a die set with rotational rolls. The multi-stage RDF process can prevent a reduction in the tooth strength of the drum clutch caused by the cut of grain flow and improve productivity via the drawing process. The rotational rolls can decrease forming load and frictional force between blank and tools. Another advantage of the multi-stage RDF process is that conventional equipment for press forming equipment can be used instead of special equipment [10].

However, multi-stage RDF process still has some problems, such as spring-back, unfilling, and elevated tool temperature during continuous manufacturing process, which mean that ensuring the dimensional accuracy of the drum clutch is difficult [10,11,12,13]. The dimensional accuracy of the final product depends on the design parameters of each process, such as the thickness of the blank, diameter of the roll, and clearance between the roll and mandrel [14,15,16]. In addition, dimensional errors can occur in subsequent stages of the multi-stage RDF process because of the elevated temperature of the tool. Therefore, an effective design method is required to achieve the dimensional accuracy of the product according to process parameters.

The purpose of this study is to design a multi-stage RDF process for the manufacturing of a drum clutch that improves dimensional accuracy using an artificial neural network (ANN). Finite element (FE) simulation of the multi-stage RDF process was performed to predict the dimensional accuracy according to various clearances for each stage. Moreover, the ANN is used to infer the relationship between the clearance and the dimensional accuracy of the drum clutch to reduce the number of FE simulations. Finally, the drum clutch was fabricated using conditions determined by the FE simulation and ANN, and the dimensional accuracy of manufactured drum clutch was compared with the results of FE simulation. 

## 2. Multi-Stage Roll Die Forming Process for Drum Clutch

### 2.1. Drum Clutch

The drum clutch, which is a main component in automotive transmission systems, performs an important role in power transmission and is the cause of much of the vibration and noise produced by an automobile. Improving the power transmission and reducing both the vibration and noise requires excellent dimensional accuracy, for example, in terms of tooth thickness, the unfilling at corners, and diameters of the drum clutch. Figure 1 shows the main dimensions and the tolerance of the drum clutch, which has two peripheral walls with different diameters.

### 2.2. Application of Multi-Stage Roll Die Forming Process

Figure 2 shows a schematic drawing of the 1st and 2nd stages of the RDF process. The inner and outer teeth of the drum clutch are formed by rolls installed within the die set during the downward movement of the mandrel and pad, similar to the case of deep drawing. The roll is rotated by the frictional force between the blank and the roll.

In this study, the multi-stage RDF process for manufacturing the drum clutch consists of two stages. The 1st stage of this process consists of three steps of RDF process to form small peripheral wall whose diameter is 154.6 mm. An initial blank is formed into the cup shape with rough tooth in the 1st step. The precise tooth shape is formed by the 2nd step, and the 3rd step is sizing process to manufacture the final product with improved dimensional accuracy. The 2nd stage of this process is to form large peripheral wall, of which diameter is 183.3 mm, and proceed subsequently by same procedure with 1st stage RDF process. Figure 3 shows the shapes of the roll and mandrel for a multi-stage RDF process for each stage.

## 3. FE Simulation of Multi-Stage Roll Die Forming Process

### 3.1. Construction of FE Model

The material used in this study was SAPH440 with a thickness of 3.6 mm. Uniaxial tensile test was performed using material test system equipment (MTS Landmark^TM^ 100 kN, MTS Systems Corporation, Eden Prairie, MN, USA) to obtain mechanical properties for application of FE simulation. Tests of SAPH440 were carried out according to ASTM E8M [17], and results are summarized in Table 1. The material model used in this study was the Hollomon equation, which is the relationship between the representative stress and strain. 

In this study, the FE simulation was performed using DEFORM 3D (Version 11, Scientific Forming Technologies Corporation, Columbus, OH, USA), with a 1/24 section of blank and dies considering the symmetry of the drum clutch. The blank was meshed with approximately 450,000 hexahedral elements, and a symmetric boundary condition at the radial edge of the blank was imposed on the FE model. Figure 4 shows the FE model for the 1st and 2nd stages of the RDF process, and the conditions for the FE simulation are listed in Table 2. The roll was rotated by the friction between the blank and the roll. In the case of RDF, the frictional force at the interface between the blank and the roll is very low because of the rolling contact of the blank with the rotational roll. In addition, actual RFD process is performed with sufficient and continuous lubrication for the roll and blank. Thus, the friction factor(m) in the RDF is assumed to be 0.02 between the roll and blank [15]. The effect of anisotropy is ignored owing to a relatively small drawing depth of 60 mm and a small plane strain during the RDF, compared with general sheet metal forming. Additionally, SAPH440 has almost isotropic properties because this blank is hot rolled steel plate with large thickness of 3.6 mm [18].

### 3.2. Results of the FE Simulation

The clearance between the mandrel and the roll during the RDF for the drum clutch is the main process parameter affecting the dimensional accuracy of the final product. In this study, the clearances are defined as design parameters, which are summarized in Table 3.

Figure 5 shows the results of the FE simulation of the dimensional accuracy with various clearances. The predicted unfilled or overfilled area ranged from 0.1419 mm^2^ to 0.2384 mm^2^ for each tooth. In the case of zero clearance, an unfilling phenomenon occurred, whereas an over-filling phenomenon was observed in the case of 10%t clearance. Figure 6 shows the deformed shape and distribution of the effective strain for case 5. The effective strain largely increases during the 1st and the 2nd step of each stage because the inner and outer teeth are almost deformed in these steps. In the 3rd step, which is used for a sizing process, the effective strain is similar to the 2nd step because amount of deformation is small. A large distribution of the effective strain is also observed for the inner teeth owing to a relatively small clearance and large deformation. This may result in an increase in the level of wear on the punch and roll dies during mass production [15].

## 4. Optimization of Multi-Stage Roll Die Forming Process

### 4.1. Artificial Neural Network

ANN is a computational model inspired by natural neurons that is applied to solve complex functions in various fields, including pattern matching, data compression, and functional approximation [19]. Multilayer perceptron trained by back propagation is the most popular and useful method used by ANNs; it consists of a number of neurons, a single hidden layer, and a nonlinear activation function. In this study, the ANN was employed to predict the dimensional accuracy of a drum clutch with a sigmoid transfer function, and it was trained to learn the nonlinear relationship between the process parameters and the dimensions of the deformed part. The back propagation algorithm was adopted to train the networks, where the dimensions predicted by FE simulation was used as the training dataset.

### 4.2. Optimization of Clearance with Artificial Neural Network

As mentioned above, the results of the FE simulation were used to train and develop the ANN. In this study, the predicted sectional area of the teeth is taken as the training input data according to clearances of each RDF stages. The output data is trained with back-propagation algorithm by ANN. In the learning process, the network is presented with an input pattern and a corresponding desired output pattern. Using the weights and thresholds, the network produces the output pattern, which is compared with the desired output pattern [20]. The results of ANN were utilized to determine the process window as shown in Figure 7. The process window is used to obtain the relationship between process parameters and dimensional accuracy and could predict dimensional accuracy for all combinations within a whole range of process parameters. In order to ensure process window, the predicted sectional area of tooth by ANN was compared with the prediction results by FE simulation. The sectional area was precisely predicted by process window in comparison with result of FE simulation within maximum error of 0.7%. Finally, the optimized clearances were determined to be 6%t for the 1st stage and 5%t for the 2nd stage for the multi-stage RDF process.

### 4.3. FE Simulation of the Manufacturing Process of the Drum Clutch with Optimized Clearance

Most manufacturing processes of gear components are designed for the initial products. However, the actual manufacturing process is continuous and repetitive. In addition, the die temperature is increased due to friction between the product and the die. Therefore, it is necessary to predict the temperature limit of the die to maximize the dimensional accuracy of the drum clutch.

An FE simulation of five cycles from the initial product to the fifth product was performed to obtain the temperature increase and distribution as shown in Figure 8. Die modeling was performed to investigate the dimensional accuracy at the maximum temperature, which was assumed to be 200 °C considering the maximum temperature of the fifth product. The predicted thickness for final product of each stage was within allowable tolerance of ±0.05 mm, as shown in Figure 9. The mechanical properties of metals were also affected by elevated temperatures. In particular, the yield strength and tensile strength decreased at elevated temperatures. Therefore, structural simulation of the die at elevated temperatures was required to predict the safety of the die.

Structural simulation was performed for the 1st and 2nd stages of RDF processes because the maximum stress applied to the die was measured during the sizing process. Figure 10 shows the results of heat transfer and forming simulation, which were used to perform the structural simulation for the rolls. The temperature distribution and the maximum load from the FE simulation of the die with a maximum temperature of 200 °C were applied to the roll. The material used for the die was AISI D2. Figure 11 shows the results of the structural simulation. The maximum stresses of 1st and 2nd stages of RDF processes are 1100 MPa and 1550 MPa, respectively. In addition, the compressive yield strength of the material is 1900 MPa [21,22]. Consequently, the plastic deformation of dies may not have occurred because the stresses on the die during both the 1st and 2nd stages were lower than their compressive yield strength. The fatigue failure could have occurred on the roll during the mass production because the maximum stress was relatively large compared with yield strength. The fatigue life of tool could be improved by prevention of stress concentration during RDF process due to rotational roll.

## 5. Experimental Verification

The experiments for multi-stage RDF process to manufacture the drum clutch were performed to verify suggested design method based on the results of FE simulation and ANN. Figure 12 shows the shapes of the mandrel and roll dies for the 2nd step of each stage. The roll and the housing were surrounded by a case to prevent the generation of gap between the housing during the multi-stage RDF process. In the RDF die set, 48 rolls were installed and tools were manufactured with AISI D2. The experiments were also performed with same conditions for FE simulations.

Figure 13 shows the results of the experiment and the FE simulation for multi-stage RDF process. The experimental results show that the tooth shape is quite sound, and excellent dimensional accuracy of product is achieved by suggested optimal clearance. Additionally, the results of the FE simulation are in good agreement with the experimentally measured dimensional accuracies within a maximum error of 2.09% as summarized in Table 4. These results showed that deformation behavior in actual process could be described well by FE simulation of multi-stage RDF process used in this study. The dimensional accuracy of the final product satisfies the target dimension within an allowable tolerance. Therefore, the experimental results demonstrate the validity of the optimal design parameters determined in this study. The process window based on FE simulation and ANN can be used efficiently for determination of process parameters in multi-stage RDF process to manufacture drum clutch.

## 6. Conclusions

In this study, a multi-stage RDF process is designed as a manufacturing method for the drum clutch. An ANN with FE simulation is performed to determine the optimal parameters and to improve the dimensional accuracy of the drum clutch. The conclusions of this study can be summarized as follows:(1)A multi-stage RDF process is suggested as a manufacturing method for the drum clutch to prevent decreases in the tooth strength and improve dimensional accuracy.(2)The clearance between the roll and the mandrel is selected as the main design parameter. An ANN with FE simulation is performed to determine the optimal parameters of the 1st and 2nd stages of the RDF process. The process window is also constructed to obtain the relationship between the process parameters and the dimensional accuracy and could predict dimensional accuracy for all combinations within a whole range of process parameters without additional FE simulation.(3)FE simulation considering a continuous multi-stage RDF confirms the feasibility of the suggested manufacturing process and parameters. The temperature increase and maximum stress were within the permitted safety tolerances of the actual manufacturing process. The fatigue life of tool could be improved by prevention of stress concentration during RDF process because of the rotational roll.(4)The optimized conditions were successfully utilized to manufacture the drum clutch. The calculated dimensional accuracy of the product was in good agreement with that of the manufactured drum clutch, and the object dimensions were within an allowable tolerance. The suggested process window can be used efficiently for determination of process parameters in multi-stage RDF process to manufacture drum clutch.

## Figures and Tables

**Figure 1 materials-14-00069-f001:**
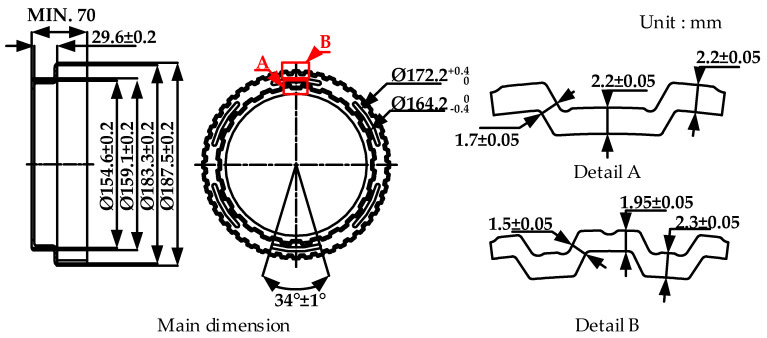
Dimensions of drum clutch.

**Figure 2 materials-14-00069-f002:**
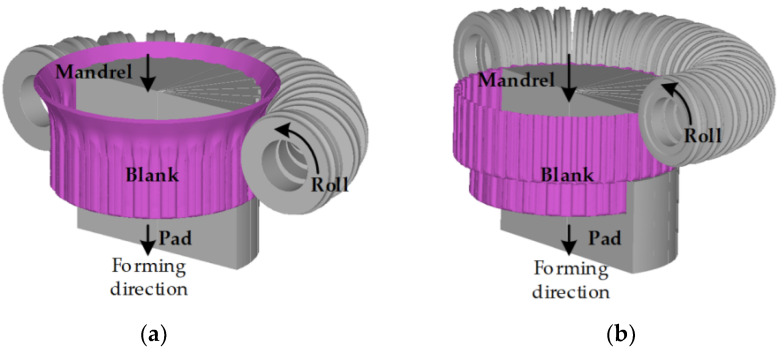
The schematic drawing multi-stage roll die forming (RDF) process, (**a**) 1st stage of RDF and (**b**) 2nd stage of RDF.

**Figure 3 materials-14-00069-f003:**
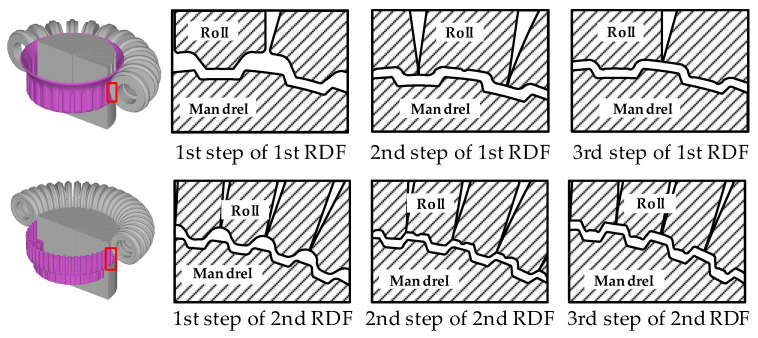
Tool shapes of the multi-stage RDF process.

**Figure 4 materials-14-00069-f004:**
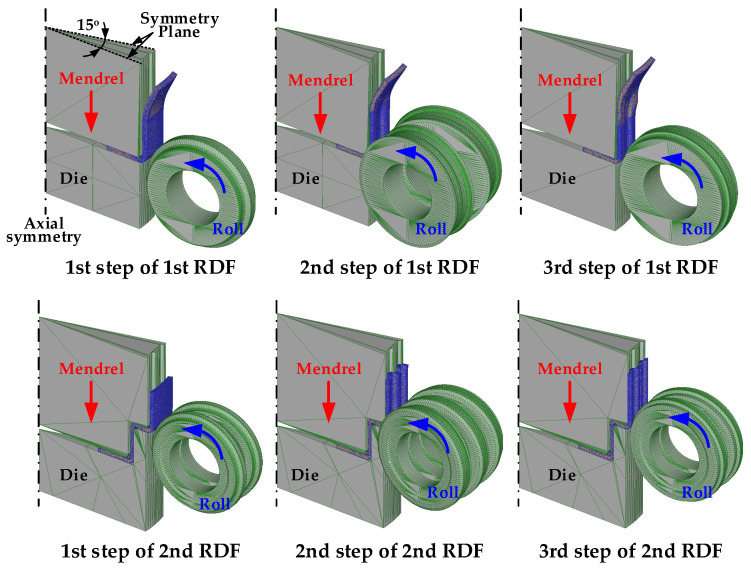
FE model for RDF process.

**Figure 5 materials-14-00069-f005:**
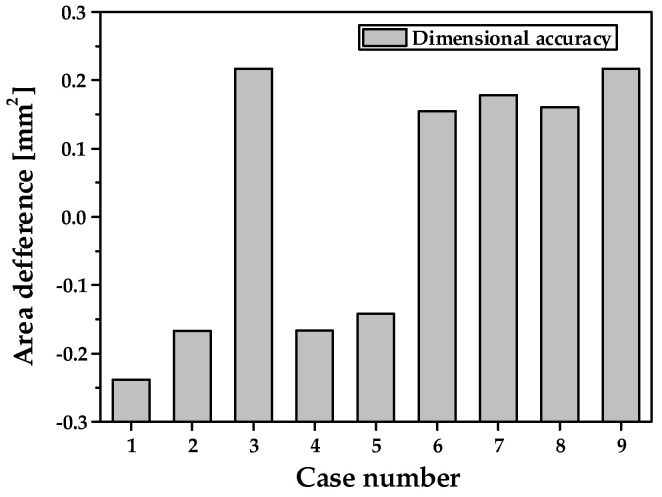
Results of the FE simulation for dimensional accuracy.

**Figure 6 materials-14-00069-f006:**
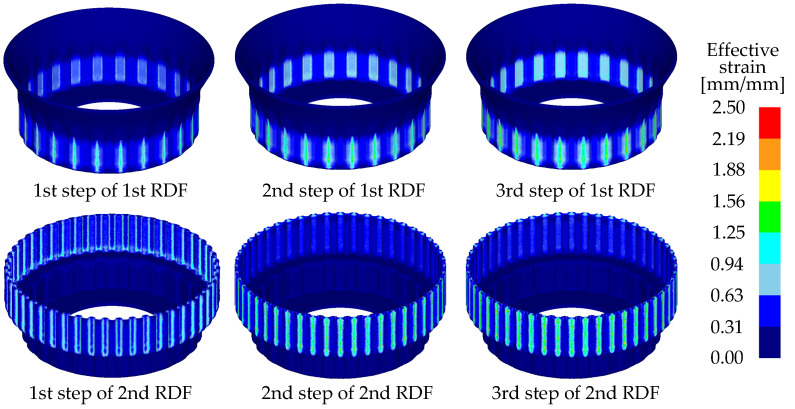
Results of the FE simulation of the deformed shape and distribution of the effective strain.

**Figure 7 materials-14-00069-f007:**
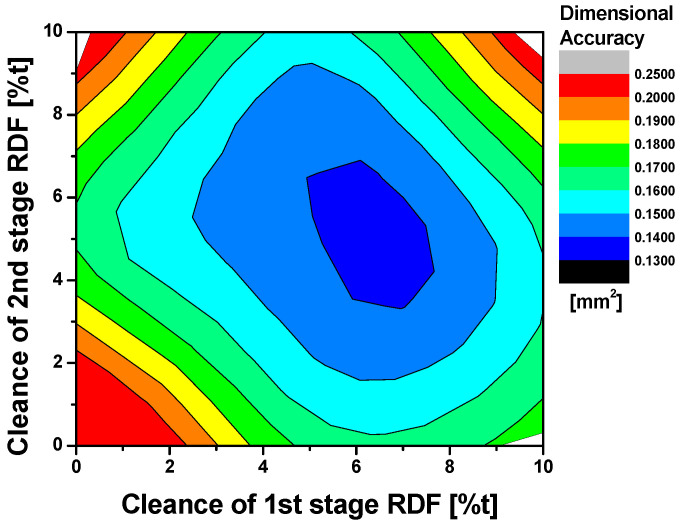
Process window to predict dimensional accuracy.

**Figure 8 materials-14-00069-f008:**
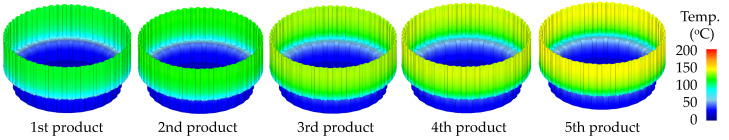
Temperature distributions from the 1st product to the 5th product.

**Figure 9 materials-14-00069-f009:**
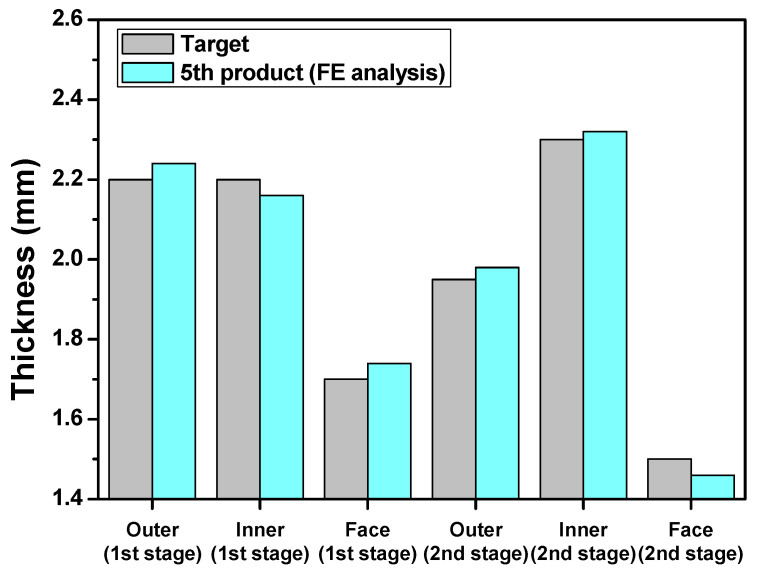
The predicted thickness for the 5th product.

**Figure 10 materials-14-00069-f010:**
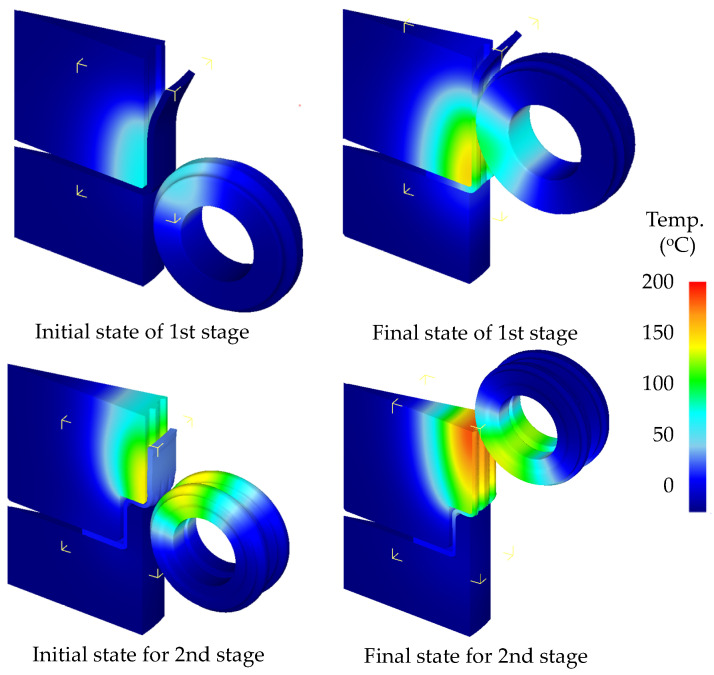
The results of heat transfer and forming simulation for 5th product.

**Figure 11 materials-14-00069-f011:**
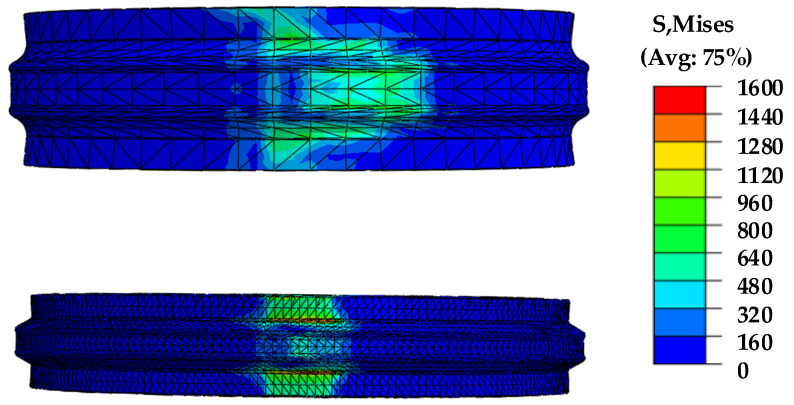
Result of structural simulation of the rolls of the 2nd RDF process.

**Figure 12 materials-14-00069-f012:**
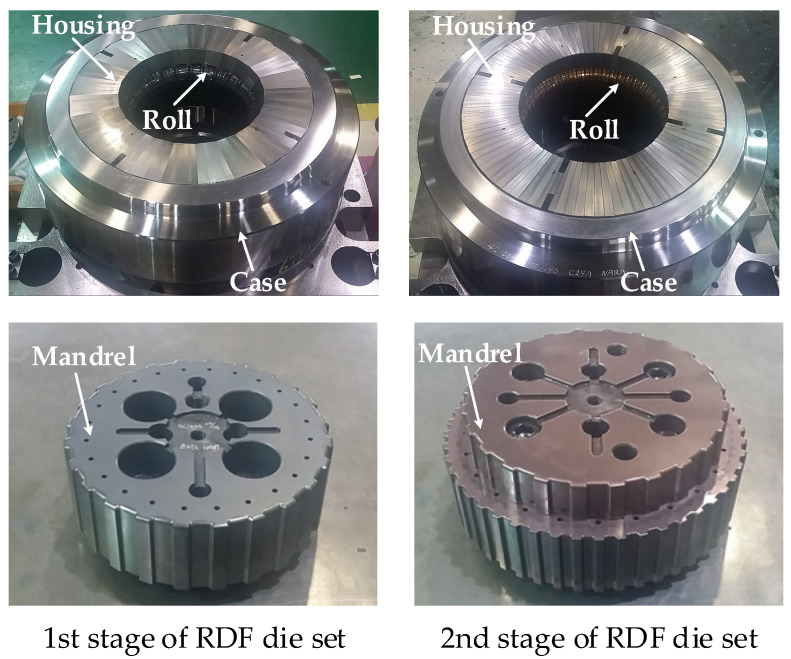
Die set for manufacturing the drum clutch.

**Figure 13 materials-14-00069-f013:**
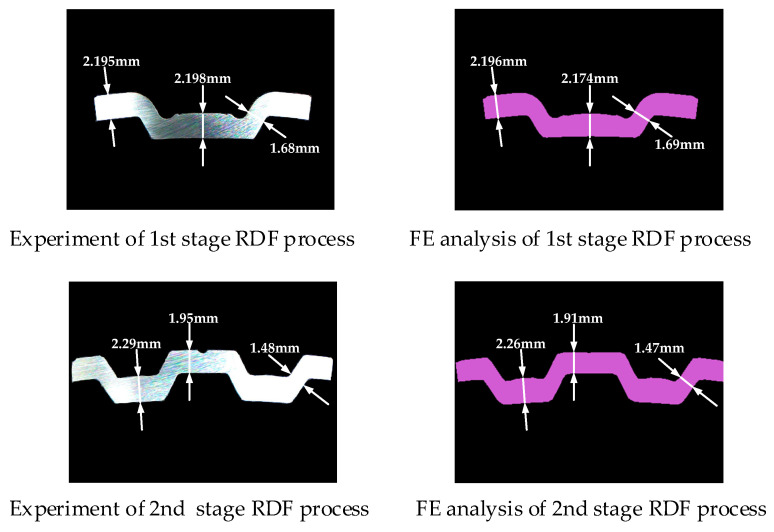
Comparison of experiment and FE simulation for tooth shape.

**Table 1 materials-14-00069-t001:** Mechanical properties for SAPH440.

Mechanical Properties	Values
Yield strength	315 MPa
Tensile strength	436 MPa
Strength coefficient (K)	722.2 MPa
Strain hardening exponent (n)	0.198
Material model (Hollomon equation)	$\sigma =722.2\cdot \varepsilon^{0.198}$

**Table 2 materials-14-00069-t002:** The conditions for the finite element (FE) simulation.

Process Condition		Values
Material		SAPH440
Punch(mandrel) speed		500 [mm/s]
Friction factor (m)	Blank with die and mandrel	0.12
	Blank with roll	0.02

**Table 3 materials-14-00069-t003:** Cases of FE simulation.

Case	Clearance of 1st Stage [%t]	Clearance of 2nd Stage [%t]	Case	Clearance of 1st Stage [%t]	Clearance of 2nd Stage [%t]
1	0	0	2	0	5
3	0	10	4	5	0
5	5	5	6	5	10
7	10	0	8	10	5
9	10	10			

**Table 4 materials-14-00069-t004:** Comparison of the experiment and FE simulation for dimensional accuracy.

		Experiment [mm]	FE Simulation [mm]	Error [%]
1st stage of RDF process	Thickness of inner tooth	2.20	2.17	1.38
	Thickness of outer tooth	2.20	2.20	0.00
	Thickness of face	1.68	1.69	0.59
	Internal diameter	154.62	154.46	0.10
	External diameter	159.19	159.08	0.07
2nd stage of RDF process	Thickness of inner tooth	2.29	2.26	1.32
	Thickness of outer tooth	1.95	1.91	2.09
	Thickness of face	1.48	1.47	0.68
	Internal diameter	183.41	183.20	0.11
	External diameter	187.65	187.44	0.11

## Data Availability

The data presented in this study are available on request from the corresponding author.

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
