# Peer review of "Design of Multi-Stage Roll Die Forming Process for Drum Clutch with Artificial Neural Network"

_materials, 2020, doi:10.3390/ma14010069_

Round 1

Reviewer 1 Report

The manuscript is focused on the design of multi-stage roll die forming process for produce drum clutch parts. In order to improve the dimensional accuracy of the drum clutch produced an artificial neural network was used. Numerical and experimental tests were performed and a good agreement between both numerical and experimental results was obtained.

The paper is interesting and fits the aim and scope of the journal.

Problems:

  • The English must be revised.
  • The references to the figures must be revised on the entire manuscript.

Introduction

  • The introduction should be revised, and new scientific articles focused on the manufacturing of drum clutch parts must be added.

 Multi-stage roll die forming process for drum clutch

  • The overall quality of the figure 1 must be improved.
  • In this chapter, the manufacturing process is explained. However, information about the transition between the 1st and the 2nd stage must be added, i.e., the die and the mandrel of the tool for the 2nd stage cause plastic deformation of the blank. This detail must be explained, and the plastic deformation must be evaluated on chapter 3.

FE simulation of multi-stage roll die forming process

  • Details about the tensile tests must be added. A good mechanical characterization is crucial in order to obtain a good correlation between experimental and numerical results.
  • Why the friction factor at the interface blank - roll is different of the friction factor at the interface blank - die/mandrel? The material of the roll is equal to the material of the die? If the frictional force at the interface blank – roll is very low, the Prandtl friction model is a good option? This information must be added in the manuscript.
  • The roll angular velocity is a consequence of the mandrel speed? Why the angular velocities are available in table 2? Table 2 is of FE conditions or conditions and results? To justify the different angular velocities of the rolls, the rolls have different outer diameters?
  • The results presented on figure 5 must be clearer. Specific area values should be represented, or positive and negative differences should be presented in order to understand the unfilling and the over-filling cases.
  • In figure 6 more steps of the manufacturing process can be added.

Optimization of multi-stage roll die forming process

  • How the maximum error of the dimensional accuracy of the FE simulations was calculated?
  • Where are the results of the FE simulations of five cycles from the initial product to the fifth product? With this FE simulations, the maximum temperature could be easily obtained.
  • The compressive yield strength of the AISI D2 material at 200ºC is 2200 MPa? A reference must be added.
  • If the compressive yield strength of the AISI D2 is approximately 2200 MPa and the maximum stress obtained on the manufacturing process is 1550 MPa, the fatigue life of the tool is relatively reduced. For this reason, information about fatigue must be added on the manuscript.
  • In figure 9, the initial conditions of the FE simulations must be presented. The initial temperature of the blank is the room temperature?

Experimental verification

  • In table 4 the data should be clearer. Inner and outer are thickness of inner tooth and thickness of outer tooth?

Reviewer 2 Report

Please check this work for english cause they are numerous phrases that are weak, examples : “the improvement of automotive transmission systems can improve” improvement -improve ???

Overall the introduction is very brief. It requires major improvements

The novelty is not very well stressed. Other wise I tend to see this paper in design or manufacturing journals as looks now “ design a multi-stage RDF process for the manufacturing”

“In this study, the multi-stage RDF process for manufacturing the drum clutch consists of two stages.” OK but which are these stages ?

Better to insert the boundary condition in Figure 4 will allows easy following this work

“relatively small drawing depth during deep drawing” please provide details of depth?

“Figure 6.” Which is the deformed one and which is undeformed ? please put caption a and b

In Figure 8 and 9 you have presented couples of images but not clear what they represent so I suggest put a caption for each one

The results should eb elaborate more and interpreted against literature. Now there are no any interpretation.

“ANN” please provide details cause is only mentioning about the ANN, but how does work and more details

In conclusion I suggest carefully improve this work that is very interesting but is very briefly presented

What about new literature ?

Round 2

Reviewer 1 Report

No comments.

Reviewer 2 Report

.